# In Vitro Verification of Simulated Daily Activities Using Implant-Specific Kinematics from In Vivo Measurements

**DOI:** 10.3390/bioengineering11111108

**Published:** 2024-11-02

**Authors:** Yashar A. Behnam, Ahilan Anantha Krishnan, Renate List, Chadd W. Clary

**Affiliations:** 1Center for Orthopaedic Biomechanics, University of Denver, Denver, CO 80210, USA; yashar.behnam@du.edu (Y.A.B.); ahilan.ananthakrishnan@du.edu (A.A.K.); 2Institute for Biomechanics, ETH Zurich, 8093 Zurich, Switzerland; renate.list@kws.ch; 3Human Performance Lab, Schulthess Clinic, 8008 Zurich, Switzerland

**Keywords:** TKA, dynamic joint simulator, knee loading boundary conditions, AMTI VIVO, activities of daily living

## Abstract

The mechanism and boundary conditions used to drive experimental joint simulators have historically adopted standardized profiles developed from healthy, non-total knee arthroplasty (TKA) patients. The purpose of this study was to use implant-specific in vivo knee kinematics to generate physiologically relevant boundary conditions used in the evaluation of cadaveric knees post-TKA. Implant-specific boundary conditions were generated by combining in vivo fluoroscopic kinematics, musculoskeletal modeling-generated quadriceps loading, and telemetric knee compressive loading during activities of daily living (ADL) to dynamically drive a servo-hydraulic knee joint simulator. Ten cadaveric knees were implanted with the same TKA components and mounted in the knee simulator to verify the resulting load profiles against reported fluoroscopic kinematics and loading captured by an ultra-congruent telemetric knee implant. The cadaveric simulations resulted in implant-specific boundary conditions, which accurately recreate the in vivo performance of the like-implanted knee, with Root Mean Square Error (RMSE) in femoral low point kinematics below 2.0 mm across multiple activities of daily living. This study demonstrates the viability of in vivo fluoroscopy as the source of relevant boundary conditions for a novel knee loading apparatus, enabling dynamic cadaveric knee loading that aligns with clinical observations to improve the preclinical development of TKA component design.

## 1. Introduction

In vitro experimental evaluations of knee joint mechanics using joint simulators are prevalent in the development of total knee arthroplasty (TKA) systems. Early versions of joint simulators were developed for implant tribological testing and used to evaluate the wear and friction behavior of TKA implants under controlled laboratory conditions [1,2,3]. Capable of applying millions of load cycles, modern commercially available joint simulators like the Bionix^®^ (MTS, Eden Prairie, MN, USA), ProSim (Simsol, Stockport, UK), and VIVO (AMTI, Watertown, MA, USA) incorporate sophisticated control systems and are considered industry standards for critical verification testing in regulatory submissions. Despite their ubiquity, questions remain about whether the implant loading conditions created by these simulators accurately reflect the in situ conditions for modern TKA systems.

The tibiofemoral (TF) loading conditions used by knee simulators rely primarily on variations of ISO-14243 [4,5,6], which specifies relative positions and loading between the tibia and femur during gait independent of the implant geometry or constraint. These loading conditions are based on historic measurements of healthy knee kinematics and kinetics using reflective markers, videography, and a force platform [7,8,9]. Notably, the subjects in these studies were not patients with TKA. Geometric features of modern TKA, like post-cam mechanisms, variable conformity between the tibia and femur, and the patellofemoral articulation fundamentally change the kinematics of the implanted knee and should be accounted for in preclinical testing [10,11,12,13,14].

More recently, strain gauges incorporated into telemetric tibial bases were implanted in patients to directly measure in vivo knee loading [15,16,17,18,19]. These measurements led to the formulation of ASTM F3141 [20], which described standardized TF implant loading for a variety of activities of daily living (ADLs), including gait, stair ascent, stair descent, rising and sitting from a chair, and a pivoting movement. A cohort of these subjects participated in a detailed kinematic assessment using mobile fluoroscopy and motion capture to directly measure both knee kinematics and kinetics during activities of daily living (i.e., CAMS knee dataset [21]). These data represent the most comprehensive measurements of implanted knee mechanics but are limited to a single highly conforming implant design (Innex^®^, ZIMMER BIOMET, Warsaw, IN, USA), and the measured loads, particularly in the axial plane, may not be directly applicable to contemporary TKA implants with lower conformity.

One challenge in applying measured TF loading conditions for knee implant evaluations is that most knee simulators do not dynamically load the knee’s extensor mechanism. In vivo TF loads are an aggregate of ground reaction forces applied through the foot, constraint from knee ligaments and capsule, and a complex array of muscle actions including the quadriceps. Applying the measured TF loads directly to the knee without the dynamic constraint provided by the extensor mechanism and knee ligaments may result in spurious knee kinematics.

### 1.1. Related Work

Whole knee joint simulators that simultaneously load the TF and patellofemoral (PF) joints are uncommon and even fewer recreate dynamic physiological loading of both joints. The Oxford knee simulator was among the first experimental rigs that utilized the knee’s extensor mechanism to counteract a compressive load applied through the hip [22,23]. Subsequent variations in the Oxford rig included hydraulic or pneumatic actuators and control systems to recreate dynamic ground reaction forces at the ankle [24,25,26] and incorporated additional actuation of muscle groups like the hamstrings [27,28,29,30]. Using Oxford-style rigs to apply standardized loading conditions is difficult because these simulators cannot directly control the loading of the TF joint, but instead apply hip and ankle loading that recruits the quadriceps muscles to approximate the desired knee loading.

Robotic knee simulators utilize robotic arms or Stewart platforms to directly apply TF loading through the tibia or femur and measure knee laxity or passive knee flexion [31,32,33,34,35]. Robotic simulators have traditionally been limited in their ability to apply dynamic unconstrained loading like the conditions experienced by the knee during activities of daily living. Recent advancements in commercial wear simulator design and control have facilitated the development of hybrid simulators that can simultaneously apply realistic PF loading through the quadriceps tendon while also applying controlled loads and displacements directly to the TF joint [36,37,38,39]. These simulators address many limitations of previous knee rigs, enabling both displacement control and dynamic load control using anatomic axes aligned to the knee. The primary challenge with leveraging these simulators is the development of the appropriate loading conditions that combine quadriceps loading with loads applied through the tibia to create the desired mechanics in the knee.

### 1.2. Study Purpose

The purpose of this study was to leverage implant-specific kinematics measured in vivo during ADLs with mobile fluoroscopy [40] to formulate boundary conditions that recreate these activities in vitro. The ADL boundary conditions were applied to a cohort of cadaveric knees implanted with the same contemporary implant system. In vitro knee kinematics were compared between the implant-specific boundary conditions and similar boundary conditions derived from the CAMS knee dataset [21]. Our hypothesis was that boundary conditions based on the CAMS knee dataset would generate implanted knee kinematics that were significantly different from the implant-specific boundary conditions and associated in vivo implanted knee kinematics. This study provides a critical assessment of the current standardized loading conditions used to evaluate the durability of TKA through a comparison with in vivo kinematic measurements. In addition, an experimental methodology to develop more realistic simulations that improve preclinical testing of future TKA was demonstrated.

## 2. Materials and Methods

### 2.1. Boundary Condition Development

A VIVO 6 degree-of-freedom (DoF) joint simulator (Advanced Mechanical Technology, Inc., Watertown, MA, USA) was retrofitted with custom fixtures and a quadriceps actuator to mount either fixtured TKA or cadaveric knee specimens into the loading rig (Figure 1). The design and verification of the simulator were described in detail in a previous publication [37]. In addition to the 6 DOF load cell housed at the base of the tibial fixture used in the control of the simulator, custom load-sensing tibial tray and patella implants were incorporated into the fixtured knee to measure TF and PF loads during testing (Figure 2, left). The instrumented tibial tray housed an array of four miniature button compression loadcells (Model LCDK-250, Omega Instruments, Norwalk, CT, USA), one in each quadrant of the tray, configured to measure superior–inferior (S-I), adduction–abduction (Ad-Ab) moment, and flexion–extension (F-E) moment applied to the tray. Calibration of the tray used a previously described methodology [18], where a calibration block with 13 steel beads in known locations was fixed to the surface of the tray, and 1000 N axial forces were applied at each bead location (Figure 2, right, Instron 8872, Norwood, MA, USA) creating a known S-I load and moments about the centroid of the tray. A calibration matrix was calculated that minimized errors between loadcell measurements and the known loading applied by the load frame.

Displacement-controlled and load-controlled boundary conditions were developed for the simulator using a combination of previously published fluoroscopically measured in vivo knee kinematics [40], compressive loading from the OrthoLoad database [41], and quadriceps forces calculated from a previous gait lab study [42]. The fluoroscopic dataset included knee F-E and anterior–posterior (A-P) translations of the lowest points on the medial and lateral condyles from 15 subjects (7 female, 8 male, age 69.2 ± 8.6 years, 18.3 ± 3.4 months postoperatively, body mass index (BMI) 27.9 ± 3.4 kg/m^2^) implanted with ATTUNE^®^ Cruciate Retaining Fixed Bearing TKA (DePuy Synthes, Warsaw, IN, USA). The subjects performed multiple cycles of gait, stair descent, sit-to-stand, and stand-to-sit movements while implant kinematics were measured using a mobile single-plane fluoroscopy system [43]. Knee low-point kinematics for each activity were averaged across all trials for a subject, normalized to the mean implant size, and then averaged across subjects. The stand-to-sit and sit-to-stand movements were concatenated to form an average sit–stand cycle. Average low-point translations were transformed into internal–external (I-E) rotations and A-P translations using the Grood and Suntay convention via a custom MatLab^®^ script (MathWorks, Natick, MA, USA). To supplement the kinematics data, tibiofemoral compressive loading during these same activities was downloaded from the OrthoLoad database for five subjects [44], normalized to the subjects’ body weight, averaged across subjects, then scaled to represent a 66 kg person (www.orthoload.com, accessed 18 May 2021). This compressive loading was used to maintain consistency with the CAMS knee dataset and ASTM F3141.

Quadriceps muscle forces for each activity were calculated from a separate previously published motion capture study of patients with the same TKA implant system. The full details of the study can be found here [42,45], so only a brief description of the study is provided. The study included a cohort of 17 subjects (average weight = 76.3 kg, 79 trials) performing gait and 18 subjects (average weight = 71.5 kg, 80 trials) performing stair descent, stand-to-sit, and sit-to-stand. Motion capture was performed using a passive marker camera system (Vicon, Centennial, CO, USA) and 4 force platforms (Bertec, Columbus, OH, USA). Kinematics and ground reaction forces were used to construct subject-specific musculoskeletal models in OpenSim [46] based on previously published methods [46,47]. Joint forces and moments were obtained using inverse dynamics and quadriceps muscle forces from static optimization that minimized the sum of muscle activations [48]. The average quadriceps forces for each patient during each activity were calculated across trials, normalized based on body weight, scaled to a 66 kg body weight person, and then averaged across subjects.

These kinematics and kinetics data were combined to formulate implant-specific loading conditions for the VIVO simulator (“Implant Specific—Displacement”, Table 1). The boundary conditions included displacement-controlled profiles for knee F-E and I-E rotations and A-P translations, while the S-I compression and quadriceps DoFs were in load control. The focus of the current analysis was axial plane knee kinematics, so medial-lateral (M-L) forces and Ad-Ab moments were assumed to be negligible and simulated with load control maintaining zero load. Using displacement control for A-P and I-E DoFs is problematic when the alignment of the specimen in the simulator cannot be accurately controlled or different implant geometry is being evaluated (e.g., implants with a post-cam mechanism). Furthermore, compliance of the test specimen can reduce the accuracy of these displacement-controlled profiles under high compressive loading. Thus, an iterative experimental process was employed to determine the I-E and A-P loading profiles necessary to generate the displacements measured from the fluoroscopic data (Figure 3).

The same mean-sized femur and insert implants used during the in vivo measurements were mounted into the VIVO simulator via the load-sensing tray and patella and synthetic bones. The “Implant Specific—Displacement” profiles were applied to the components via the simulator’s integrated control system. Implant kinematics during the simulations were measured with an optical tracking system (OPTOTRAK Certus HD, Northern Digital, Waterloo, ON, Canada) to account for compliance in the experimental fixtures. The A-P forces and I-E moments applied by the simulator’s tibial actuator were recorded and then subsequently used as input force profiles for the same DoFs. To compensate for compliance in the fixturing, the A-P and I-E loading profiles were iteratively modified until the desired low-point kinematics were attained (root mean square error (RMSE) ≤ 2.0-mm). The resulting A-P and I-E loading profiles were recorded for the “Implant Specific—Load” boundary condition (Table 1, Figure 4).

The CAMS-Knee dataset represents the most comprehensive combined measurements of knee kinetics and knee kinematics available and has been extensively used to evaluate TKA mechanics [21]. However, it remains unclear whether the measured loading, particularly in the transverse plane, is appropriate to evaluate contemporary TKA systems with more moderate articular conformity. For this reason, alternative gait, stair descent, and sit–stand boundary conditions were developed from the CAMS-Knee dataset to facilitate a comparison with the Implant-Specific profiles. In the CAMS-Knee dataset, 6 subjects (1 female, 5 male, aged 74 ± 5 years, mass 89 ± 13 kg, height 172 ± 4 cm) were implanted with Zimmer INNEX^®^ TKA retrofit with telemetric sensors to measure 6-DoF TF reaction forces. This implant system has sagittal and axial TF conformities of 0.99 throughout the flexion range, which is higher than many modern TKA systems. Patients performed the activities of daily living while implant kinematics were measured using mobile fluoroscopy, along with motion capture and telemetric TF loading. Knee F-E was calculated for each trial from the motion capture markers. TF loading profiles for each activity were averaged across all trials for each subject, normalized to body weight, averaged across subjects, and then scaled to a 66 kg body weight. The average F-E rotations were combined with the average TF loading and the same quadriceps force profile from the “Implant-Specific” boundary conditions to formulate the “CAMS” boundary conditions (Table 1, Figure 4).

### 2.2. In Vitro Kinematics Assessment

Twenty fresh frozen cadaveric knees (10 specimens, 8M, 2F, avg. age = 81, avg. BMI = 23.8) were implanted with the same TKA system used during the boundary condition development by fellowship-trained board-certified orthopedic surgeons using a diverse set of surgical philosophies (Appendix A). Specimens were screened for lower extremity trauma, prior surgery, cancer, chronic smoking, and BMI > 40. Prior to surgery, high-resolution computed tomography scans were performed through the length of the femur and tibia. During surgery, surgeons utilized a mid-vastus approach to preserve the quadriceps tendon. The manufacturer’s surgical technique was followed, with the posterior tibial slope matching the native plateau. After surgery, the femur and tibia were transected mid-shaft with care taken to preserve the knee’s soft tissue and quadriceps tendon. The proximal femur and distal tibia were skeletonized and then cemented into cylindrical fixtures with polymethyl methacrylate (bone cement). Each knee was mounted into the AMTI VIVO and adjusted so that the femoral implant was aligned to the simulator’s femoral F-E and Ad-Ab axes (Appendix A).

The suite of knee boundary conditions, including the implant-specific (load control) and CAMS variations of gait, stair descent, and sit–stand were applied to the knees. Three cycles of each boundary condition were performed while knee kinematics were measured during the second cycle. Knee kinematics were measured with active-marker arrays attached to the femur and tibia fixtures and patella bone using the optical tracking camera system described earlier. Fiducial markers were attached to the bone during testing and probed in the local bony coordinate systems to enable registration of the bony anatomy and implants after surgery. After testing, the knees were skeletonized and an optical scanner was used to digitize the bone and implant geometry, along with the fiducial markers (Space Spider, Artec3D, L2328, Luxembourg). Femur, tibia, and patella bone geometry reconstructed from the CT scan and CAD models of the implants were fit to the digitized geometry using an iterative closest point algorithm, then registered to the experimental kinematics using the fiducial markers. Femur, tibia, and patella local implant coordinate systems were established relative to the implant geometry, and Grood and Suntay knee kinematics were calculated for each cycle. Likewise, the lowest points on the femur articular geometry were identified relative to the plane of the tibial tray. Kinematic analysis was conducted using custom MatLab^®^ scripts (MathWorks, Natick, MA, USA).

Knee kinematics across specimens were synchronized using the most extended knee flexion at the beginning of the loading cycle. Gait and stair descent activities were divided into stance and swing phases with heel strike at 0% of the cycle length and toe-off at 60%. The mean and standard deviation of the knee kinematics across specimens were calculated. RMSE between the in vivo fluoroscopically measured knee kinematics and corresponding kinematics of the cadaveric knees during the same activities was also calculated.

To detect statistical differences between the in vivo and in vitro time-series kinematics, one-dimensional statistical parametric mapping (SPM with 2-tailed *t*-tests, v0.4, www.spm1d.org) was implemented in Python [49]. Assuming a Gaussian distribution of the time-series data, SPM implements random field theory to make statistical inferences through time without dimensional reduction or information loss [50]. The null hypothesis was that there were no differences between the in vivo kinematics and the in vitro kinematics during corresponding activities. A post hoc Bonferroni correction was used to adjust the acceptance of the null hypothesis with an alpha level of 0.05.

## 3. Results

The results of this study include a detailed comparison of the implant-specific loading conditions derived from the in vivo fluoroscopic data with loading measured using the CAMS-Knee dataset (Section 3.1) and the resulting knee kinematics when both loading regimes were applied to a cohort of cadaveric knees (Section 3.2).

### 3.1. Implant-Specific Knee Boundary Condition Development

Following the iterative process of tuning the displacement-controlled implant-specific profiles on fixtured components, the RMSE of the LP A-P translations was lowest during gait for the medial condyle (0.7 mm) and during sit–stand for the lateral condyle (0.7 mm). The average RMSE values of the femoral A-P translations for both condyles during all activities were 1.2-mm and 1.3-mm, respectively.

Notable differences were observed in the A-P and I-E loading for the implant-specific and CAMS boundary conditions across activities (Figure 4). For the stance phase of gait, the implant-specific A-P loading experienced two loading peaks in the posterior direction; the first just after heel strike (−71 N) and the second just prior to toe-off (−101 N). In contrast, the CAMS profile prescribed a single posterior peak load of −125 N during mid-stance. The I-E torque during stance for the implant-specific loading exhibited a single external peak (3.1 Nm) while CAMS oscillated between an external peak in the early stance (−2.0 Nm) and an internal peak in the late stance (3.7 Nm). Both sets of boundary conditions exhibited minimal A-P or I-E loading during the swing phase of gait (after a 60% gait cycle). Quadriceps loading, which was the same for both gait profiles, peaked at 651 N in early stance with a second peak of 271 N to initiate the swing phase of gait. Tibiofemoral compressive loading and patella loading measured using the instrumented implants for the implant-specific boundary conditions are reported in Figure 5. The applied quadriceps load during gait generated a peak PF load of 468 N in early stance, combined with the applied compression by the simulator to generate a peak TF compressive load of 1798 N at 17% of the gait cycle.

During stair descent, both implant-specific and CAMS boundary conditions prescribed a posterior force on the tibia, which peaked at −172 N for the implant-specific and −129 N for CAMS loading profiles (Figure 4, middle row). The implant-specific loading presented a second posterior peak of −136 N just prior to toe-off when the CAMS A-P loading was minimal. The implant-specific stair descent I-E loading exhibited two internal peaks, one near mid-stance (4.2 Nm) and a second larger peak near toe-off (5.0 Nm). The CAMS loading profiles had minimal I-E torques until a 2.7 Nm internal moment occurred during toe-off. Quadriceps loading during both profiles peaked near heel strike (645 N), again in mid-stance (582 N), and a final time at toe-off (251 N, Figure 5). The patella was not in contact with the femur until mid-stance due to knee extension. The peak PF reaction force of 853 N coincided with the second peak of the quadriceps loading. Like gait, the quadriceps loading increased the peak TF compressive load to 2187 N in the early stance.

For the sit–stand activity, the posteriorly directed reaction force for the implant-specific loading increased with increasing flexion, reaching a maximum of −167 N at peak flexion (Figure 4, bottom). This posterior force was accompanied by a large internal torque that also increased with flexion to 10.4 Nm. In contrast, the CAMS A-P loading profile switched from −52 N posterior in extension to 35 N anterior at maximum flexion, accompanied by a much smaller internal torque peaking at 3.6 Nm at the start of the standing portion of the cycle. It should be noted that the CAMS knee compressive load dropped from 1372 N to only 490 N in peak flexion after contact with the chair, while the implant-specific profile assumed contact with the chair did not occur. The quadriceps load peaked at 660 N during maximum flexion, resulting in a PF resultant force of 1405 N and a TF compressive load of 1933 N in the implant-specific loading conditions.

The PF loading was decomposed into components along the articular surface based on the anatomic coordinate system of the patella. The largest component of the PF reaction force acted in the anterior direction at 375 N during gait, 615 N during stair descent, and 1300 N during sit-to-stand. During all three activities, the patella experienced a medially directed load acting on the articular surface that increased with knee flexion, reaching a maximum of 80 N for both stair descent and deep knee bend. The loading on the patella was generally oriented superiorly for gait (100 N), stair descent (175 N), and stand-to-sit (130 N). However, the reaction vector briefly shifted inferiorly (−40 N) at the initiation of standing from the seated position. The measured PF loading for all implant-specific activities is listed in Appendix A.

### 3.2. Evaluation of Boundary Conditions in Cadaveric Knees

The RMSE between the fluoroscopy-measured in vivo condylar low-point A-P translations and the cadaveric condylar translations using the implant-specific boundary conditions ranged between 0.8 mm (medial condyle during sit–stand) and 2.3 mm (lateral condyle during gait, Table 2). RMSE for the CAMS boundary conditions ranged from 2.6 mm (medial condyle during gait, lateral condyle during sit–stand) to 5.1 mm (medial condyle during stair descent, Table 3). The implant-specific loading conditions had a lower RMSE than the CAMS loading conditions for the low-point translations of both condyles in all three activities. Similarly, the implant-specific boundary conditions had lower RMSE in A-P and I-E knee kinematics, except for I-E rotation during the sit–stand cycle, where stance and swing phase of the implant-specific boundary exceeded 2° during stand–sit and sit–stand phases (Table 2 and Table 3).

During gait, the in vivo fluoroscopy exhibited minimal A-P translations of the medial femoral condyle low-point during stance, with anterior oscillation occurring during the swing phase (Figure 6 and Figure 7, top). The lateral condyle articulated slightly posterior to the medial condyle during most of the stance with the same anterior oscillation during the swing, resulting in a moderate internal translation of the tibia relative to the femur throughout the gait cycle. Despite the low RMSE between the in vivo and cadaveric low-point kinematics with the implant-specific boundary conditions for both the medial (1.1 mm) and lateral (2.3 mm) condyles, statistically significant differences in the low-point kinematics were observed during mid-stance and mid-swing for both medial (19% of cycle) and lateral (40% of cycle) condyles, primarily due to the small standard deviations observed in vivo (Appendix A). In contrast, the CAMS gait loading condition resulted in low point kinematics that were significantly different from the in vivo kinematics over 71% of the cycle for the medial condyle and 67% of the cycle for the lateral condyle.

During stair descent, both medial and lateral condyles experienced minimal A-P translations (i.e., <±5.5 mm) in vivo throughout the duration of the activity (Figure 6 and Figure 7, middle). The implant-specific cadaveric stair descent kinematics resulted in similar kinematics, with RMSE less than 1.8 mm for the duration of the cycle (Table 2). However, the resulting low-point kinematics were statistically significantly different over much of the cycle (51% of the cycle for the medial condyle and 41% for the lateral condyle). In contrast, the CAMS stair-descent loading resulted in a large anterior translation of the knee from the terminal stance through the swing phase that significantly departed from the in vivo kinematics over most of the cycle (85% of the cycle for the medial condyle and 92% for the lateral condyle). Despite the differences in the A-P position of the knee, both cadaveric profiles recreated the I-E rotation of the knee for most of the cycle.

During sit–stand, the in vivo kinematics were 7.1° of external femoral rotation with flexion, generated by a 2.1 mm anterior translation of the medial condyle and a lateral condyle, which tracked consistently 4 mm posterior to the insert dwell (Figure 6 and Figure 7, bottom). The implant-specific sit–stand loading conditions recreated the in vivo medial condyle position over 89% of the cycle with an RMSE of 0.8 mm. The lateral condyle, however, exhibited an increased posterior rollback in flexion, causing increased external femoral rotation during the sit-to-stand portion of the cycle, which was statistically different from the in vivo position over 60% of the cycle. The CAM sit–stand cycle resulted in similar levels of external rotation as the in vivo knee kinematics but also exhibited a statistically larger 6.4 mm anterior translation of the medial condyle with flexion. This was coupled with a more anterior position of the lateral condyle in the extended knee. As a result, the medial condyle position was significantly more anterior than the in vivo knee position over 80% of the cycle.

## 4. Discussion

The objective of this study was to develop implant-specific force-controlled boundary conditions derived directly from in vivo fluoroscopic analysis of implant kinematics that incorporate loading of the extensor mechanism and PF joint. These implant-specific boundary conditions were applied to a cohort of cadaveric knees using a novel servo-hydraulic testing rig with a force-controlled actuator representing the quadriceps muscle. The results of the cadaveric simulations demonstrated that the implant-specific boundary conditions accurately recreated the in vivo kinematics of the implanted knee, with RMSE in femoral low-point kinematics ranging between 0.8 mm and 2.0 mm across multiple activities of daily living. To the authors’ knowledge, this is the first attempt to develop robust force-controlled implant-specific loading conditions for the evaluation of whole knee mechanics.

### 4.1. Boundary Conditions for Simulation of Knee Mechanics

Due to their central role in the research and development of TKA, the boundary conditions used to facilitate biomechanical simulations of TKA have been scrutinized, but without consensus. Previous comparisons between treadmill gait kinematics of TKA patients and the original ISO force-controlled boundary conditions in ISO 14243-1 and the displacement-controlled boundary conditions in ISO 14243-3 demonstrated significant differences in the resulting knee kinematics [51,52,53,54]. Specifically, TKA patients in vivo exhibited significantly larger femoral A-P translations and I-E rotations during both stance and swing phases of gait than prescribed by the standard. It should be noted these studies used optical motion capture which does not directly measure implant kinematics and is influenced by soft tissue artifacts [55]. An analogous study measuring treadmill TKA gait with single-plane fluoroscopy found similar kinematic trends to the ISO profile [53], but this study had a limited sample size. Circa 2014, after the publication of these studies, the displacement-controlled ISO standard was updated to reverse the directionality of the A-P and I-E kinematics without substantive rationale or detailed comparison to contemporary in vivo kinematic data [9,54].

The mobile-fluoroscopic data used to derive the current implant-specific boundary conditions addressed some limitations of previous studies, specifically measuring overground gait versus treadmill gait, using a higher-accuracy mobile fluoroscope to measure full gait cycles instead of lower-accuracy motion capture techniques, and inclusion of additional activities. Previous comparisons of treadmill to overground walking have shown lower ground reaction forces and moments in treadmill gait [56] and faster cadence, smaller stride length, and a reduction in joint angles specifically for elderly subjects [57]. The fluoroscopic kinematics and associated cadaveric simulations using the implant-specific boundary conditions generated knee kinematics, unlike either version of the ISO standard. Specifically, the A-P translation of the medial condyle was small (range of motion (ROM) < 5.5 mm), with a general trend of posterior to anterior sliding during stance, like the data reported by DesJardins et al. using the original ISO standard [53]. The current fluoroscopic data also demonstrated an anterior sliding of the femur during the swing phase that returned to neutral translation prior to heel strike, most like the updated ISO standard [9], but this pattern was attenuated in the cadaveric simulations. While standardized boundary conditions provide a consistent method to compare TKA devices, particularly in tribological studies, there is little evidence that they accurately reflect implant mechanics in a way that enables the development of new implants to address the current challenges in TKA (i.e., fixation and instability) [58]. The results of the current study demonstrate that implant-specific boundary conditions should be used, when possible, to accurately recreate the in vivo operating conditions of the implants studied.

As previously discussed, the implant used to generate the CAMS-Knee dataset is ultra-congruent through the flexion range and is not representative of many contemporary knee implants. Given the higher level of conformity, we hypothesized that the TF loading generated by the ultra-congruent implant would be higher than more moderately conforming designs. This hypothesis was supported by recent modeling work that showed ultra-congruent TKA caused higher interface stresses at bone–implant interfaces [59,60]. When applied to the cadaveric knees, the CAM loading conditions indeed generated A-P ranges of motion that were significantly larger than the in vivo knee kinematics, particularly for the medial condyle. For gait, stair descent, and sit–stand, the A-P ROM for the medial condyle during CAMs loading was 7.5 mm, 11.4 mm, and 9.3 mm, respectively, compared to 2.7 mm, 6.4 mm, and 3.7 mm for the implant-specific loading conditions. However, the absolute peak A-P force and I-E torque in the implant-specific boundary conditions were higher than the CAM loading profiles during both stair descent and sit–stand activities. This indicates significant interactions of the applied TF loading with the knee loading generated by the extensor mechanism. Interestingly, the A-P and I-E loading profiles in the implant-specific gait simulation were most like the original ISO 14243-1 loading profiles. Both loading profiles had two posteriorly directed peaks near heal strike and toe-off and a single peak of internal torque occurring through terminal stance phase gait, although the magnitudes of those peaks were considerably smaller in the implant-specific gait profiles (Figure 8).

### 4.2. Contribution of PF Loading to Knee Mechanics

The incorporation of a dynamic extensor mechanism into the current simulation introduced an element of complexity beyond previous experimental simulations. By applying loads directly to the TF joint via actuators attached to the tibia and femur, and independently loading the PF joint through the quadriceps tendon, the simulator used in this study addressed the control challenges commonly faced by traditional Oxford knee rigs, which apply hip and ankle loads to approximate the desired knee loading [61]. Previous verification testing of the current simulator demonstrated the effects of PF loading on TF kinematics and kinetics [37]. Specifically, patella alta-baja influenced extensor efficiency, with patella alta resulting in more efficient load transfer through the patella and into the TF interface (i.e., higher TF compressive loading) through the flexion range [26,62]. The implant-specific boundary conditions were developed using a fixtured patella with a nominal patella height (Blackburne Peel (BP) ratio = 0.8) [63]. While care was taken to restore the native joint line during the cadaveric surgeries, natural variation in the length of the patellar tendon led to variation in the BP ratio across specimens (mean BP = 0.81, range = 0.6 to 1.22, Appendix A). BP ratios less than 0.5 and greater than 1.0 are considered abnormal. Using this standard, 19 of 20 knees tested in this study were within the normal range.

### 4.3. Kinematic Variability

The fluoroscopic kinematics used to derive the implant-specific boundary conditions exhibited minimal variation across study subjects, with small standard deviations in femoral low-point translations ranging from 1.1 mm to 1.5 mm for the medial condyle across activities, particularly when compared to the magnitude of low-point translations. Variation in the condylar translations between subjects was consistently lower for the medial condyle (St. Dev. = 1.2 mm) compared to the lateral condyle (St. Dev. = 1.9 mm). The standard deviation of the implant-specific cadaveric condylar translations was moderately higher for the cadaveric knees, with standard deviations of 2.0 mm and 2.1 mm for medial and lateral condyles, respectively. The variation was noticeably higher during the swing phase of gait, where low TF compressive loads reduced the constraint provided by the articulation enabling larger condylar excursions with minimal transverse plane loading.

TKA kinematics are known to vary significantly between subjects and are influenced by implant alignment, implant geometry, patient anatomy, and ligament balancing [64,65]. The patients enrolled in the fluoroscopic study were recruited from three clinics, with surgeries performed by the five surgeons in the practice. The cadaveric surgeries were performed by seven different fellowship-trained orthopedic surgeons using a variety of surgical techniques (Appendix A). Variations in technique included mechanical alignment with 3° external femoral rotation (measured resection, 8 knees), mechanical alignment with balanced flexion gaps (5 knees), patient-specific alignment with neutral femoral rotation (3 knees), and patient-specific alignment with balanced flexion gaps (4 knees). Furthermore, the posterior cruciate ligament was retained in 12 specimens and sacrificed in the remaining 8 specimens. The decision to use a diverse set of surgical techniques was intentional to assess the robustness of the force-controlled implant-specific boundary conditions across variations in surgical technique. Preliminary comparisons between cohorts of knees with each surgical condition showed no significant differences in kinematics, so all knees were grouped together in a single cohort for this study. Future work will include a detailed analysis of the relationships between implant alignment and knee kinematics across a variety of implant designs, although this analysis was beyond the scope of the current study.

### 4.4. General Applicability of Methods

The implant-specific boundary conditions derived in this study are reported in tabular form in Appendix A. These boundary conditions, derived for a contemporary moderately conformity cruciate-retaining TKA system, provide a useful comparison for experimental simulators and computational models that incorporate an extensor mechanism in simulations of activities of daily living. The applicability of these boundary conditions to implant systems with different articular surfaces should be carried out with caution. Given that the boundary conditions utilize load control for all degrees of freedom besides knee flexion, these boundary conditions should be robust to small changes in articular constraint. Additional work is required to determine whether using these same boundary conditions for different implant systems will result in physiological kinematics, including additional fluoroscopic measurements and cadaveric experimentation of alternative implant designs. However, the methodology to develop implant-specific boundary conditions demonstrated here should be considered when in vivo kinematics data are available for the implant being evaluated rather than using standardized knee loading.

### 4.5. Limitations and Future Work

This study had several limitations regarding simplifications made to facilitate the development of the implant-specific boundary conditions. First, the implant-specific profiles were developed in fixtured components without soft tissue and with a synthetic extensor mechanism. Ideally, implant-specific boundary conditions would have been developed on cadaveric knees implanted with the same TKA. However, the time-consuming iterative process of profile optimization made this untenable within the current experimental constraints. The moderately conforming nature of the articulation and relatively small A-P and I-E knee movements observed during the fluoroscopic measurements likely prevented extensive recruitment of soft tissue to stabilize the knee, minimizing the differences between fixtured and cadaveric configurations. Second, the TF compression loading profiles were not compensated to account for the compressive forces generated by the applied quadriceps load. The TF compression profiles were based on measurements by OrthoLoad trays [41], which included contributions from ground reaction forces, soft tissue, and the extensor mechanism. As evidenced by the TF compressive forces measured by the instrument tibial tray in this study, the TF compression felt by the insert during profile development was considerably higher than measured by OrthoLoad (Figure 5). The increased articular constraint from the higher compressive loading likely contributed to the larger A-P and I-E loading in the implant-specific stair descent and sit–stand simulations. Finally, the quadriceps loading, TF compressive loading, and TF kinematics used to formulate the implant-specific boundary conditions were from different sources. While the fluoroscopic study used to calculate kinematics and the gait lab study used to calculate quadriceps loading used the same implant system and activities, the patients were different. Ideally comprehensive future studies, like the CAMS-Knee study, can be performed with implant geometries representative of contemporary implant designs to facilitate implant-specific and patient-specific profile development.

Many of these limitations will be addressed through future work. Specifically, the extensive cadaveric data collected in this study will be used to develop a family of specimen-specific FE models with tuned ligaments and extensor mechanisms that can be used in implant development. These models will enable further refinement of the experimental simulations to account for the contributions of implant alignment, ligament tension, and the extensor mechanism to knee kinematics. Likewise, this same experimental method will be used to assess the role of implant design, including variations in articular constraints, on knee kinematics and stability. While it remains unproven that the boundary conditions developed in this study are generalizable across implant designs with similar features, future in vivo studies should be conducted to demonstrate the predictive ability of these methods when used during the development process. In general, there is a paucity of comprehensive fluoroscopic studies for new devices introduced into clinical use due to cost, although they are becoming more common with recent advancements in image processing [66,67,68]. Ideally, experimental methods like those described here can be used to better understand implant kinematics of new devices prior to clinical use.

## 5. Conclusions

The results of this study highlight how in vivo fluoroscopy of TKA patients can be used to generate physiologically relevant boundary conditions in a novel experimental apparatus used for knee joint loading. The experimental method proved capable of dynamically loading whole cadaveric knees in a manner consistent with clinical observations. These improved boundary conditions will facilitate future investigations into the current challenges in TKA, including assessing the influence of surgical techniques and implant design on instability and implant fixation.

## Figures and Tables

**Figure 1 bioengineering-11-01108-f001:**
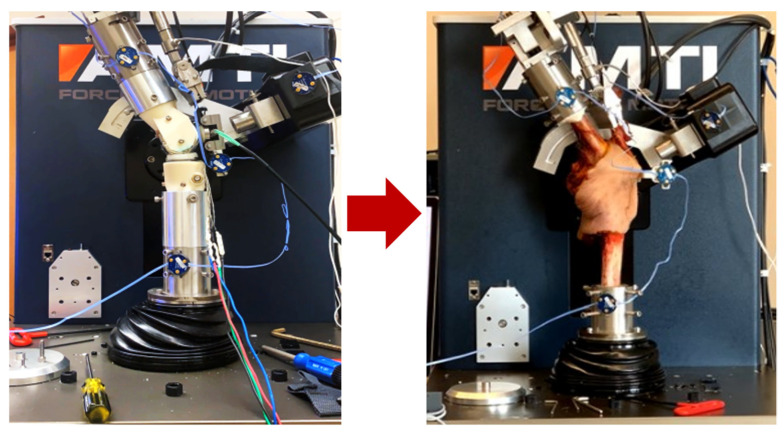
The knee simulator with fixtured implants, instrumentation, and quadriceps actuator (**left**), and the knee simulator with an intact cadaveric knee (**right**).

**Figure 2 bioengineering-11-01108-f002:**
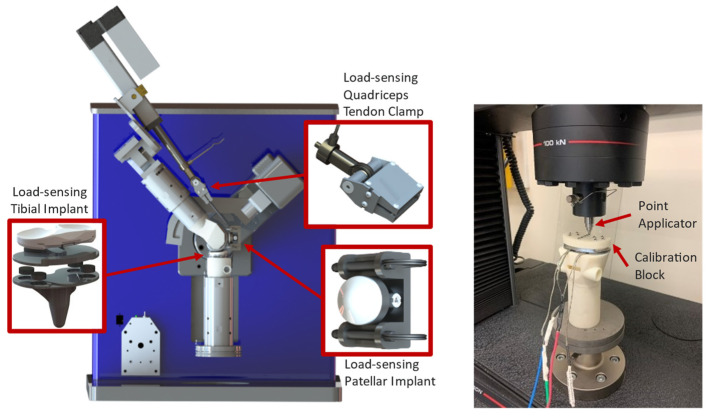
AMTI VIVO retrofit with custom fixturing, capable of loading synthetic or cadaveric knees in 6 DoF with simulated quadriceps actuation, and load-sensing tibial and patellar implants and quadriceps tendon clamp (**left**). The load-sensing tibial component loaded into a uniaxial test frame for calibration (**right**).

**Figure 3 bioengineering-11-01108-f003:**
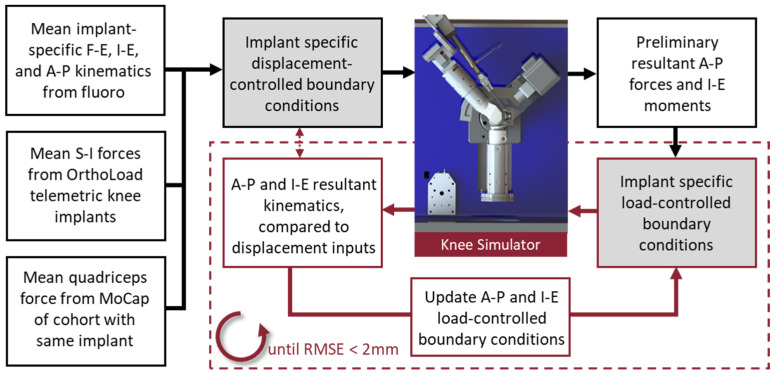
Schematic illustrating conversion of the in vivo fluoroscopically measured kinematics into load-controlled boundary conditions for the knee simulator.

**Figure 4 bioengineering-11-01108-f004:**
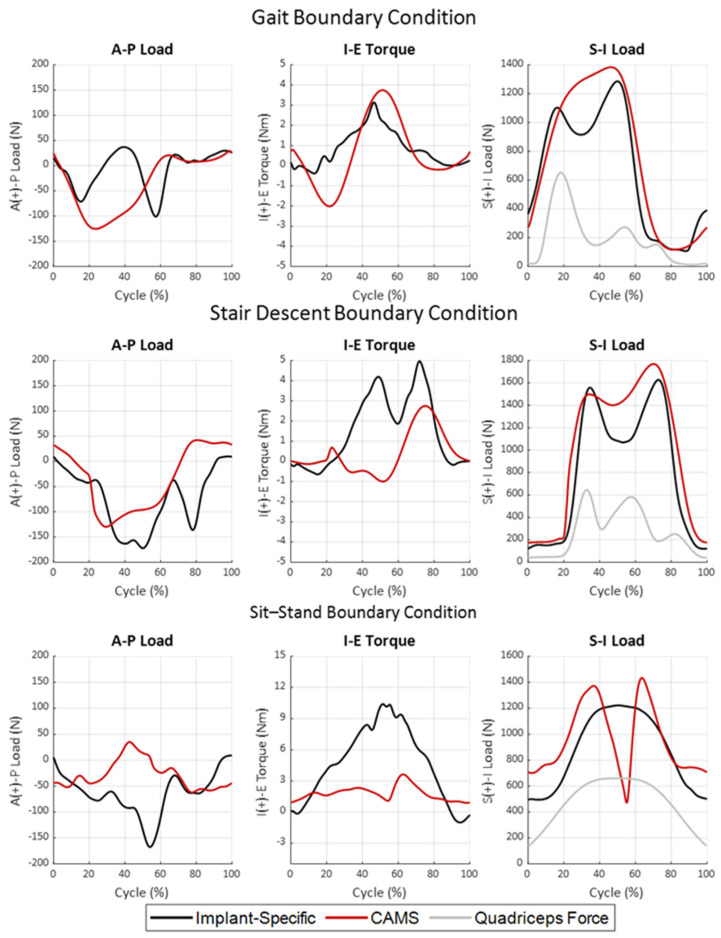
Dynamic TF A-P and TF I-E loading derived from in vivo fluoroscopic knee kinematics [40] and in vivo telemetric implants [19], as well as dynamic quadriceps loading derived from gait lab measurements and musculoskeletal modeling [42] during gait (**top row**), stair descent (**middle row**), and sit–stand (**bottom row**) activities of daily living.

**Figure 5 bioengineering-11-01108-f005:**
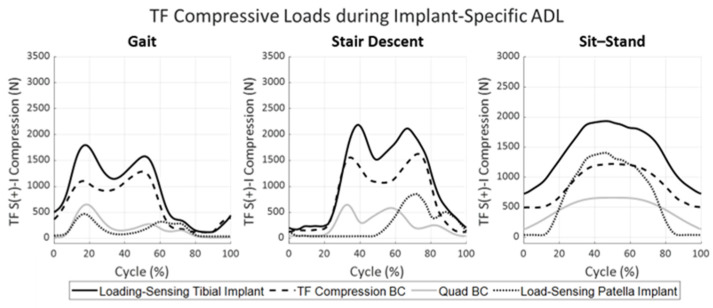
TF compression and PF loads observed at the knee articular surface in synthetic bones. TF compression was applied through the base of the tibia by the joint simulator (dashed) and quadriceps force was applied by the quadriceps actuator (gray). TF and PF resultant forces were measured by the load-sensing tibial (bold) and patella components (dotted) during the implant-specific ADL loading conditions.

**Figure 6 bioengineering-11-01108-f006:**
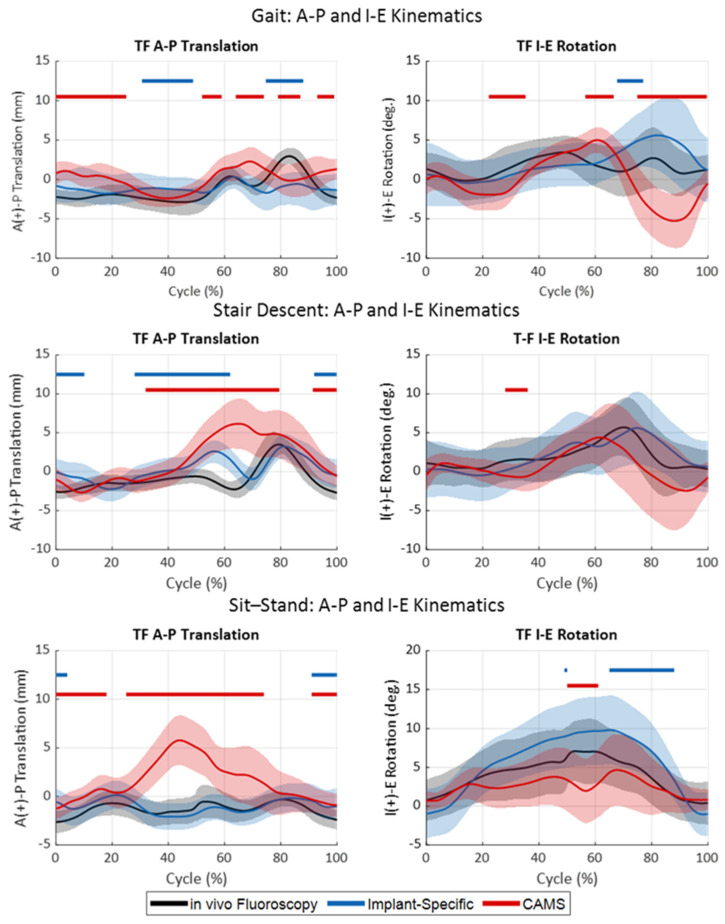
TF kinematics during gait (**top**), stair descent (**middle**), and sit–stand (**bottom**) activities. In vivo kinematics (black) were compared to the kinematics of the cadaveric specimen using either the implant-specific boundary condition (blue) or CAMS-Knee boundary conditions (red). Shaded regions indicated one standard deviation. Horizontal bars indicate regions of significant difference between the in vivo kinematics and the two sets of cadaveric kinematics.

**Figure 7 bioengineering-11-01108-f007:**
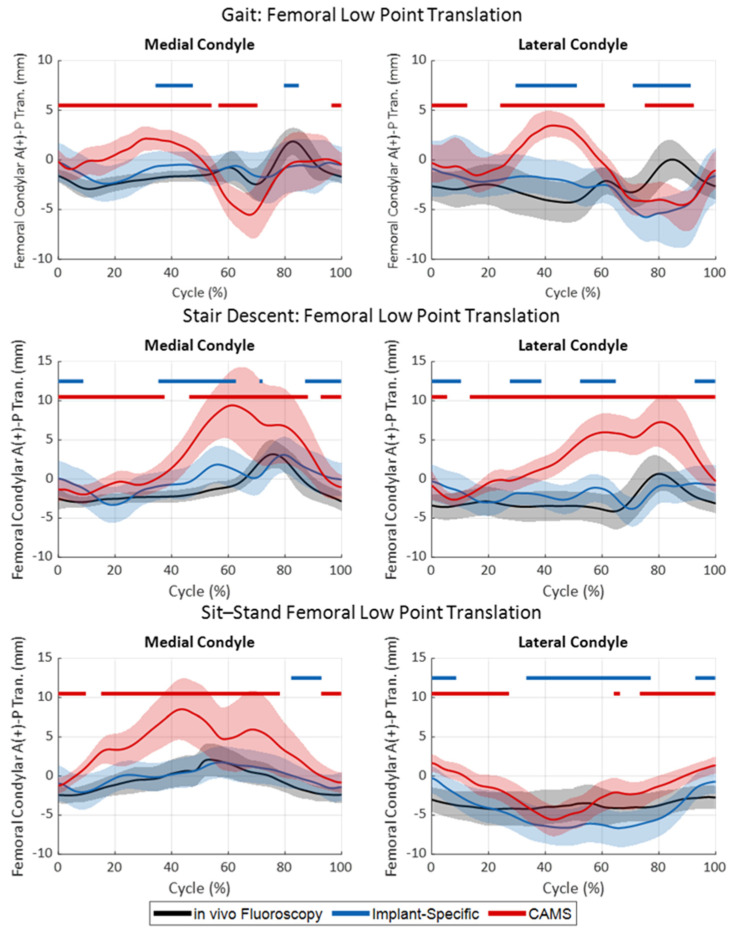
Femoral condylar low point A-P translations (medial condyle, left, and lateral condyle, right) during gait (**top**), stair descent (**middle**), and sit–stand (**bottom**) activities. In vivo kinematics (black) were compared to the kinematics of the cadaveric specimen using either the implant-specific boundary condition (blue) or CAMS-Knee boundary conditions (red). Shaded regions indicated one standard deviation. Horizontal bars indicate regions of significant difference between the in vivo kinematics and the two sets of cadaveric kinematics.

**Figure 8 bioengineering-11-01108-f008:**
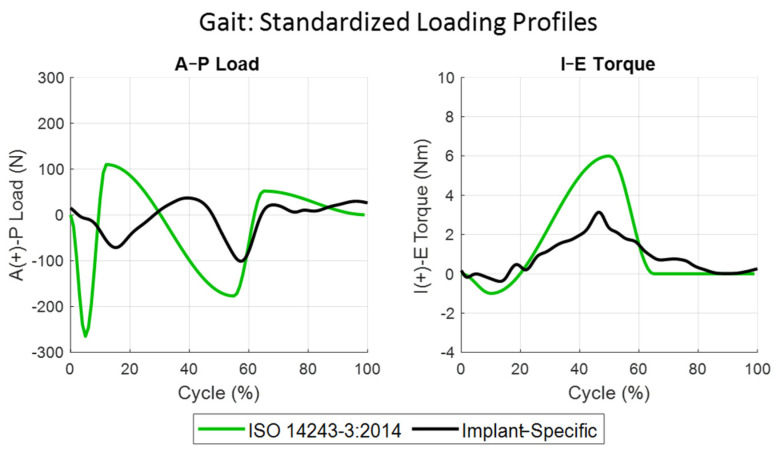
ISO 14243-3:2014 (green) standardized force-control profile for the loading of knee prosthesis during the gait activity compared to the implant-specific (black) profile developed in this study.

**Table 1 bioengineering-11-01108-t001:** Boundary conditions developed for use with the VIVO simulator. “DoF” indicates the controllable degrees of freedom of the VIVO simulator (displacement or load control). “Source” indicates the data source from which the profile was created. Sources include implant-specific fluoroscopy (fluoro), the OrthoLoad database (OrthoLoad), motion capture analysis (MoCap), or the CAMS-Knee data repository (CAMS). Degrees of freedom that were maintained with zero force or moment are also indicated.

	Implant Specific—Displacement		Implant Specific—Load		CAMS-Knee	
DoF	Control Mode	Source	Control Mode	Source	Control Mode	Source
F-E	Displacement	Fluoro	Displacement	Fluoro	Displacement	CAMS
Ad-Ab	Load	0 Moment	Load	0 Moment	Load	0 Moment
I-E	Displacement	Fluoro	Load	Derived	Load	CAMS
M-L	Load	0 Force	Load	0 Force	Load	0 Force
A-P	Displacement	Fluoro	Load	Derived	Load	CAMS
S-I	Load	OrthoLoad	Load	OrthoLoad	Load	OrthoLoad
Quad.	Load	MoCap	Load	MoCap	Load	MoCap

**Table 2 bioengineering-11-01108-t002:** Root mean square error (RMSE) between reported in vivo fluoroscopy and cadaveric kinematics for femoral low-point A-P translation and TF kinematics under the implant-specific boundary condition.

	Femoral Low-Point A-P Translation						TF Kinematics					
	Medial Condyle (mm)			Lateral Condyle (mm)			A-P Translation (mm)			I-E Rotation (deg)		
Activity	Cycle	Stance	Swing	Cycle	Stance	Swing	Cycle	Stance	Swing	Cycle	Stance	Swing
Gait	1.1	0.9	1.3	2.3	1.5	3.0	1.5	1.1	1.9	1.8	1.0	2.6
Stair Descent	1.8	1.8	1.8	1.7	1.7	1.7	1.8	1.7	2.0	1.2	0.8	1.5
Activity	Cycle	Stand-Sit	Sit-Stand	Cycle	Stand–Sit	Sit–Stand	Cycle	Stand–Sit	Sit–Stand	Cycle	Stand–Sit	Sit–Stand
Sit–Stand	0.8	0.6	0.9	1.9	1.8	2.0	0.7	0.8	0.6	2.4	2.0	2.7

**Table 3 bioengineering-11-01108-t003:** Root mean square error (RMSE) between in vivo fluoroscopic and the cadaveric CAMS femoral low-point kinematics and TF kinematics.

	Femoral Low-Point A-P Translation						TF Kinematics					
	Medial Condyle (mm)			Lateral Condyle (mm)			A-P Translation (mm)			I-E Rotation (deg)		
Activity	Cycle	Stance	Swing	Cycle	Stance	Swing	Cycle	Stance	Swing	Cycle	Stance	Swing
Gait	2.6	2.8	2.2	3.9	4.7	2.5	2.8	3.3	1.7	3.2	1.6	4.6
Stair Descent	5.1	4.6	5.6	4.0	3.6	4.5	4.0	3.0	5.1	1.8	1.1	2.5
Activity	Cycle	Stand–Sit	Sit–Stand	Cycle	Stand–Sit	Sit–Stand	Cycle	Stand–Sit	Sit–Stand	Cycle	Stand–Sit	Sit–Stand
Sit–Stand	4.7	5.4	4.0	2.6	2.8	2.4	2.2	2.5	2.0	2.0	1.6	2.4

## Data Availability

The datasets generated and supporting the findings of this article are obtainable from the corresponding author upon reasonable request.

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
