# Peer review of "In Vitro Verification of Simulated Daily Activities Using Implant-Specific Kinematics from In Vivo Measurements"

_bioengineering, 2024, doi:10.3390/bioengineering11111108_

Round 1
Reviewer 1 Report
Comments and Suggestions for Authors
This study proposed implant-specific force-controlled boundary conditions from in vivo fluoroscopic analysis of implant kinematics that incorporates the extensor mechanism and PF joint loading.
I shall suggest the following enhancement:
1. The authors must state the contribution of their work in the introduction.
2. The authors should include the existing state-of-the-art techniques in the Related Work section, categorize them into broader groups based on their underlying algorithms, and discuss their pros and cons.
3. Explain the reason for including existing work (Figure 4).
4. The authors shall present their results more clearly.
Minor editing of English language required.
Reviewer 2 Report
Comments and Suggestions for Authors
This study aims to leverage implant-specific kinematics measured in vivo during ADLs with mobile fluoroscopy to formulate boundary conditions that recreate these activities in vitro. This reviewer finds this paper to be highly interesting and impactful. The manuscript is also well-structured and presents sufficient results to support the validity of the research. However, This reviewer recommends that the authors include a discussion on the general applicability of the proposed method in the discussion section.
Round 2
Reviewer 1 Report
Comments and Suggestions for Authors
Since the authors address all the comments, I would like to accept this paper.
Comments on the Quality of English LanguageMinor editing of English language required.